# Automatic Neuron Detection in Calcium Imaging Data Using Convolutional Networks

**Noah J. Apthorpe**[1]*    **Alexander J. Riordan**[2]*    **Rob E. Aguilar**[1]    **Jan Homann**[2]
**Yi Gu**[2]    **David W. Tank**[2]    **H. Sebastian Seung**[12]
[1]Computer Science Department    [2]Princeton Neuroscience Institute
Princeton University
{apthorpe, ariordan, dwtank, sseung}@princeton.edu

## Abstract

Calcium imaging is an important technique for monitoring the activity of thousands of neurons simultaneously. As calcium imaging datasets grow in size, automated detection of individual neurons is becoming important. Here we apply a supervised learning approach to this problem and show that convolutional networks can achieve near-human accuracy and superhuman speed. Accuracy is superior to the popular PCA/ICA method based on precision and recall relative to ground truth annotation by a human expert. These results suggest that convolutional networks are an efficient and flexible tool for the analysis of large-scale calcium imaging data.

## 1   Introduction

Two-photon calcium imaging is a powerful technique for monitoring the activity of thousands of individual neurons simultaneously in awake, behaving animals [1, 2]. Action potentials cause transient changes in the intracellular concentration of calcium ions. Such changes are detected by observing the fluorescence of calcium indicator molecules, typically using two-photon microscopy in the mammalian brain [3]. Repeatedly scanning a single image plane yields a time series of 2D images. This is effectively a video in which neurons blink whenever they are active [4, 5].

In the traditional workflow for extracting neural activities from the video, a human expert manually annotates regions of interest (ROIs) corresponding to individual neurons [5, 1, 2]. Within each ROI, pixel values are summed for each frame of the video, which yields the calcium signal of the corresponding neuron versus time. A subsequent step may deconvolve the temporal filtering of the intracellular calcium dynamics for an estimate of neural activity with better time resolution. The traditional workflow has the deficiency that manual annotation becomes laborious and time-consuming for very large datasets. Furthermore, manual annotation does not de-mix the signals from spatially overlapping neurons.

Unsupervised basis learning methods (PCA/ICA [6], CNMF [7], dictionary learning [8], and sparse space-time deconvolution [9]) express the video as a time-varying superposition of basis images. The basis images play a similar role as ROIs in the traditional workflow, and their time-varying coefficients are intended to correspond to neural activities. While basis learning methods are useful for finding active neurons, they do not detect low-activity cells—making these methods inappropriate for studies involving neurons that may be temporarily inactive depending on context or learning [10].

Such subtle difficulties may explain the lasting popularity of manual annotation. At first glance, the videos produced by calcium imaging seem simple (neurons blinking on and off). Yet automating image analysis has not been trivial. One difficulty is that images are corrupted by noise and artifacts due to brain motion. Another difficulty is variability in the appearance of cell bodies, which vary

in shape, size, spacing, and resting-level fluorescence. Additionally, different neuroscience studies may require differing ROI selection criteria. Some may require only cell bodies [5, 11], while others involve dendrites [6]. Some may require only active cells, while others necessitate both active and inactive cells [10]. Some neuroscientists may wish to reject slightly out-of-focus neurons. For all of these reasons, a neuroscientist may spend hours or days tuning the parameters of nominally automated methods, or may never succeed in finding a set of parameters that produces satisfactory results.

As a way of dealing with these difficulties, we focus here on a supervised learning approach to automated ROI detection. An automated ROI detector could be used to replace manual ROI detection by a human expert in the traditional workflow, or could be used to make the basis learning algorithms more reliable by providing good initial conditions for basis images. However, the usability of an automated algorithm strongly depends on it attaining high accuracy. A supervised learning method can adapt to different ROI selection criteria and generalize them to new datasets. Supervised learning has become the dominant approach for attaining high accuracy in many computer vision problems [12].

We assemble ground truth datasets consisting of calcium imaging videos along with ROIs drawn by human experts and employ a precision-recall formalism for quantifying accuracy. We train a sliding window convolutional network (ConvNet) to take a calcium video as input and output a 2D image that matches the human-drawn ROIs as well as possible. The ConvNet achieves near-human accuracy and exceeds that of PCA/ICA [6].

The prior work most similar to ours used supervised learning based on boosting with hand-designed features [13]. Other previous attempts to automate ROI detection did not employ supervised machine learning. For example, hand-designed filtering operations [14] and normalized cuts [15] were applied to image pixel correlations.

The major cost of supervised learning is the human effort required to create the training set. As a rough guide, our results suggest that on the order of 10 hours of effort or 1000 annotated cells are sufficient to yield a ConvNet with usable accuracy. This initial time investment, however, is more than repaid by the speed of a ConvNet at classifying new data. Furthermore, the marginal effort required to create a training set is essentially zero for those neuroscientists who already have annotated data. Neuroscientists can also agree to use the same trained ConvNets for uniformity of ROI selection across labs.

From the deep learning perspective, an interesting aspect of our work is that a ConvNet that processes a spatiotemporal (2+1)D image is trained using only spatial (2D) annotations. Full spatiotemporal annotations (spatial locations and times of activation) would have been more laborious to collect. The use of purely spatial annotations is possible because the neurons in our videos are stationary (apart from motion artifacts). This makes our task simpler than other applications of ConvNets to video processing [16].

## 2 Neuron detection benchmark

We use a precision-recall framework to quantify accuracy of neuron detection. Predicted ROIs are classified as false positives (FP), false negatives (FN), and true positives (TP) relative to ground truth ROIs. Precision and recall are defined by

$$\text{precision} = \frac{TP}{TP \,+\, FP} \qquad \text{recall} = \frac{TP}{TP \,+\, FN} \qquad (1)$$

Both measures would be equal to 1 if predictions were perfectly accurate, i.e. higher numbers are better. If a single measure of accuracy is required, we use the harmonic mean of precision and recall, $1/F_1 = (1/\text{precision} + 1/\text{recall})/2$. The $F_1$ score favors neither precision nor recall, but in practice a neuroscientist may care more about one measure than the other. For example, some neuroscientists may be satisfied if the algorithm fails to detect many neurons (low recall) so long as it produces few false positives (high precision). Other neuroscientists may want the algorithm to find as many neurons as possible (high recall) even if there are many false positives (low precision).

For computing precision and recall, it is helpful to define the *overlap* between two ROIs $R_1$ and $R_2$ as the Jaccard similarity coefficient $|R_1 \cap R_2|/|R_1 \cup R_2|$ where $|R|$ denotes the number of pixels in $R$. For each predicted ROI, we find the ground truth ROI with maximal overlap. The ground truth ROIs with overlap greater than 0.5 are assigned to the predicted ROIs with which they overlap the most. These assignments are true positives. Leftover ROIs are the false positives and false negatives.

We prefer the precision-recall framework over the receiver operating characteristic (ROC), which was previously used as a quantitative measure of neuron detection accuracy [13]. This is because precision and recall do not depend on true negatives, which are less well-defined. (The ROC depends on true negatives through the false positive rate.)

**Ground truth generation by human annotation** The quantitative measures of accuracy proposed above depend on the existence of ground truth. For the vast majority of calcium imaging datasets, no objectively defined ground truth exists, and we must rely on subjective evaluation by human experts. For a dataset with low noise in which the desired ROIs are cell bodies, human experts are typically confident about most of their ROIs, though some are borderline cases that may be ambiguous. Therefore our measures of accuracy should be able to distinguish between algorithms that differ widely in their performance but may not be adequate to distinguish between algorithms that are very similar.

Two-photon calcium imaging data were gathered from both the primary visual cortex (V1) and medial entorhinal cortex (MEC) from awake-behaving mice (Supplementary Methods). All experiments were performed according to the Guide for the Care and Use of Laboratory Animals, and procedures were approved by Princeton University's Animal Care and Use Committee.

Each time series of calcium images was corrected for motion artifacts (Supplementary Methods), average-pooled over time with stride 167, and then max-pooled over time with stride 6. This downsampling in time was arbitrarily chosen to reduce noise and make the dataset into a more manageable size. Human experts then annotated ROIs using the ImageJ Cell Magic Wand Tool [17], which automatically generates a region of interest (ROI) based on a single mouse click. The human experts found 4006 neurons in the V1 dataset with an average of 148 neurons per image series and 538 neurons in the MEC dataset with an average of 54 neurons per image series.

Human experts used the following criteria to select neurons: 1. the soma was in the focal plane of the image—apparent as a light doughnut-like ring (the soma cytosol) surrounding a dark area (the nucleus), or 2. the area showed significantly changing brightness distinguishable from background and had the same general size and shape expected from a neuron in the given brain region.

After motion correction, downsampling, and human labeling, the V1 dataset consisted of 27 16-bit grayscale multi-page TIFF image series ranging from 28 to 142 frames per series with $512 \times 512$ pixels per frame. The MEC dataset consisted of 10 image series ranging from 5 to 28 frames in the same format. Human annotation time was estimated at one hour per image series for the V1 dataset and 40 minutes per images series for the MEC dataset. Each human-labeled ROI was represented as a $512 \times 512$ pixel binary mask.

# 3   Convolutional network

**Preprocessing of images and ground truth ROIs.** Microscopy image series from the V1 and MEC datasets were preprocessed prior to network training (Figure 1). Image contrast was enhanced by clipping all pixel values above the 99th percentile and below the 3rd percentile. Pixel values were then normalized to $[0, 1]$. We divided the V1 series into 60% training, 20% validation, and 20% test sets and the MEC series into 50% training, 20% validation, and 30% test sets.

Neighboring ground truth ROIs often touched or even overlapped with each other. For the purpose of ConvNet training, we shrank the ground truth ROIs by replacing each one with a 4-pixel radius disk located at the centroid of the ROI. The shrinkage was intended to encourage the ConvNets to separate neighboring neurons.

**Convolutional network architecture and training.** The architecture of the (2+1)D ConvNet is depicted in Figure 2. The input is an image stack containing $T$ time slices. There are four convolutional layers, a max pooling over all time slices, and then two pixelwise fully connected layers. This yields two 2D grayscale images as output, which together represent the softmax probability of each pixel being inside an ROI centroid.

The convolutional layers were chosen to contain only 2D kernels, because the temporal downsampling used in the preprocessing (§2) caused most neural activity to last for only a single time frame. Each output pixel depended on a $37 \times 37 \times T$ pixel field of view in the input, where $T$ is the number of frames in the input image stack—governed by the length of the imaging experiment and the imaging

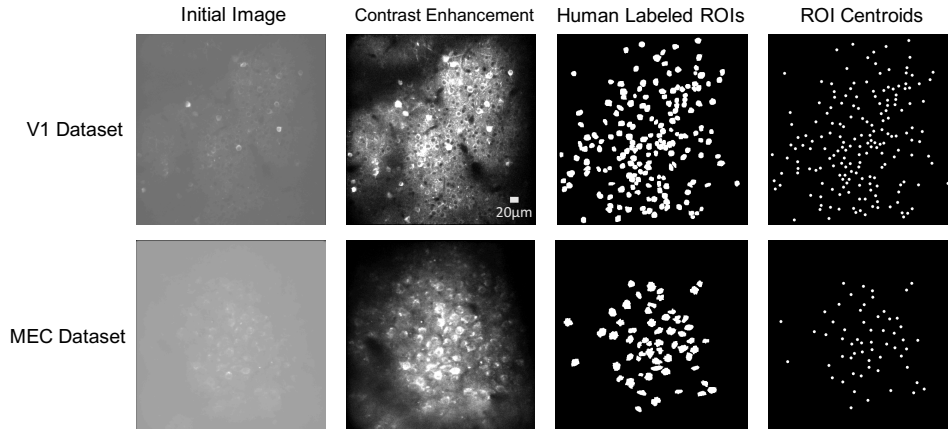

Figure 1: Preprocessing steps for calcium images and human-labeled ROIs. **Col 1**) Calcium imaging stacks were motion-corrected and downsampled in time. **Col 2**) Image contrast was enhanced by clipping pixel intensities below the 3rd and above the 99th percentile then linearly rescaling pixel intensities between these new bounds. **Col 3**) Human-labeled ROIs were converted into binary masks. **Col 4**) Networks were trained to detect 4-pixel radius circular centroids of human-labeled ROIs. Primary visual cortex (V1, **Row 1**) and medial entorhinal cortex (MEC, **Row 2**) datasets were preprocessed identically.

sampling rate. $T$ was equalized to 50 for all image stacks in the V1 dataset and 5 for all image stacks in the MEC dataset using averaging and bicubic interpolation. In the future, we will consider less temporal downsampling and the use of 3D kernels in the convolutional layers.

The ConvNet was applied in a $37 \times 37 \times T$ window, sliding in two dimensions over the input image stack to produce an output pixel for every location of the window fully contained within the image bounds. For comparison, we also trained a 2D ConvNet that took as input the time-averaged image stack and did no temporal computation (Figure 2).

We used ZNN, an open-source sliding window ConvNet package with multi-core CPU parallelism and FFT-based convolution [18]. ZNN automatically augmented training sets by random rotations (multiples of 90 degrees) and reflections of image patches to facilitate ConvNet learning of invariances. The training sets were also rebalanced by the fraction of pixels in human-labeled ROIs to the total number of pixels. See Supplementary Methods for further details.

The (2+1)D network was trained with softmax loss and output patches of size $120 \times 120$. The learning rate parameter was annealed by hand from 0.01 to 0.002, and the momentum parameter was annealed by hand from 0.9 to 0.5. The network was trained for 16800 stochastic gradient descent (SGD) updates for the V1 dataset, which took approximately 1.2 seconds/update ($\sim 5.5$hrs) on an Amazon EC2 c4.8xlarge instance (Supplementary Figure 1). The network was trained for 200000 SGD updates for the MEC dataset, which took approximately 0.1 seconds/update ($\sim 5.5$hrs).

The 2D network training omitted annealing of the learning rate and momentum parameters. The 2D network was trained for 14000 SGD updates for the V1 dataset, which took approximately 0.9 seconds/update ($\sim 3.75$hrs) on an Amazon EC2 c4.8xlarge instance (Supplementary Figure 1). We performed early stopping on the network after 10200 SGD updates based on the validation loss.

**Network output postprocessing.** Network outputs were converted into individual ROIs by: 1. Thresholding out pixels with low probability values, 2. Removing small connected components, 3. Weighting resulting pixels with a normalized distance transform, 4. Performing marker-based watershed labeling with local max markers, 5. Merging small watershed regions, and 6. Automatically applying the ImageJ Cell Magic Wand tool to the original images at the centroids of the watershed regions. Thresholding and minimum size values were optimized using the validation sets (Supplementary Methods).

**Source code.** A ready-to-use pipeline, including pre- and postprocessing, ConvNet training, and precision-recall scoring, will be publicly available for community use (`https://github.com/NoahApthorpe/ConvnetCellDetection`).

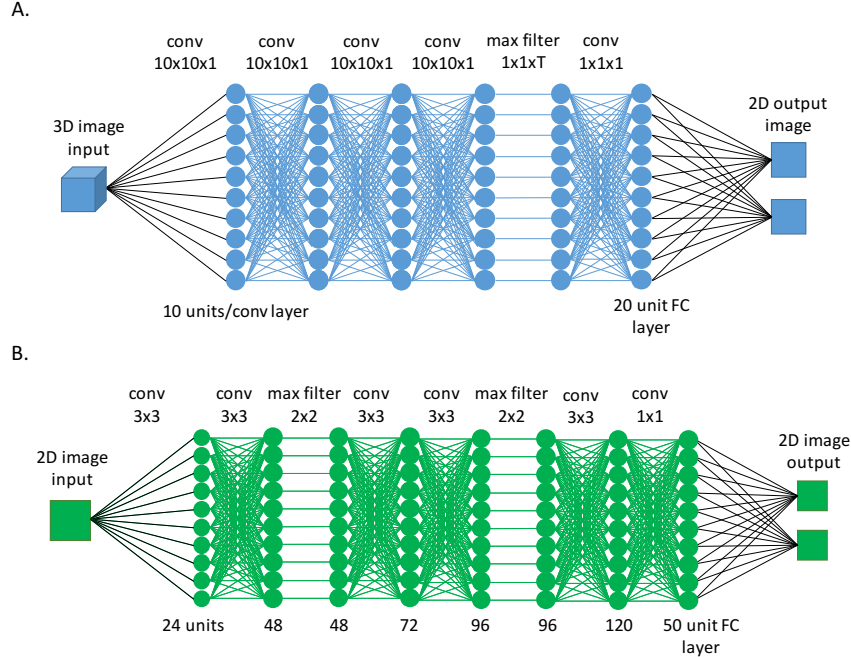

Figure 2: **A**) Schematic of the (2+1)D network architecture. The (2+1)D network transforms 3D calcium imaging stacks – stacks of 2D calcium images changing over time – into 2D images of predicted neuron locations. All convolutional filters are 2D except for the 1x1xT max filter layer, where T is the number of frames in the image stack. **B**) The 2D network architecture. The 2D network takes as input calcium imaging stacks that are mean projected over time down to two dimensions.

## 4 Results

**ConvNets successfully detect cells in calcium images.** A sample image from the V1 test set and ConvNet output is shown in Figure 4. Postprocessing of the ConvNet output yielded predicted ROIs, many of which are the same as the human ROIs (Figure 4c). As described in Section 2, we quantified agreement between ConvNet and human using the precision-recall formalism. Both (2+1)D and 2D networks attained the same $F_1$ score (0.71). Full precision-recall curves are given in Supplementary Figure 1.

Inspection of the ConvNet-human disagreements suggested that some were not actually ConvNet errors. To investigate this hypothesis, the original human expert reevaluated all disagreements with the (2+1)D network. After reevaluation, 131 false positives became true positives, and 30 false negatives became true negatives (Figure 4D). Some of these reversals appeared to involve unambiguous human errors in the original annotation, while others were ambiguous cases (Figure 4E–G). After reevaluation, the $F_1$ score of the (2+1)D network increased to 0.82. The $F_1$ score of the human expert's reevaluation relative to his original annotation was 0.89. These results indicate that the ConvNet is nearing human performance.

**(2+1)D versus 2D network.** The (2+1)D and 2D networks achieved similar precision, recall, and $F_1$ scores on the V1 dataset; however, the (2+1)D network produced raw output with less noise than the 2D network (Figure 3). Qualitative inspection also indicates that the (2+1)D network finds transiently active and transiently in focus neurons missed by the 2D network (Figure 3). Although such neurons occurred infrequently in the V1 dataset and did not noticeably affect network scores, these results suggest that datasets with larger populations of transiently active or variably focused cells will particularly benefit from (2+1)D network architectures.

**ConvNet segmentation outperforms PCA/ICA.** The (2+1)D network was also able to successfully locate neurons in the MEC dataset (Figure 5). For comparison, we also implemented and applied PCA/ICA as described by Ref. [6]. The (2+1)D network achieved an $F_1$ score of 0.51, while PCA/ICA achieved 0.27. Precision and recall numbers are given in Figure 5.

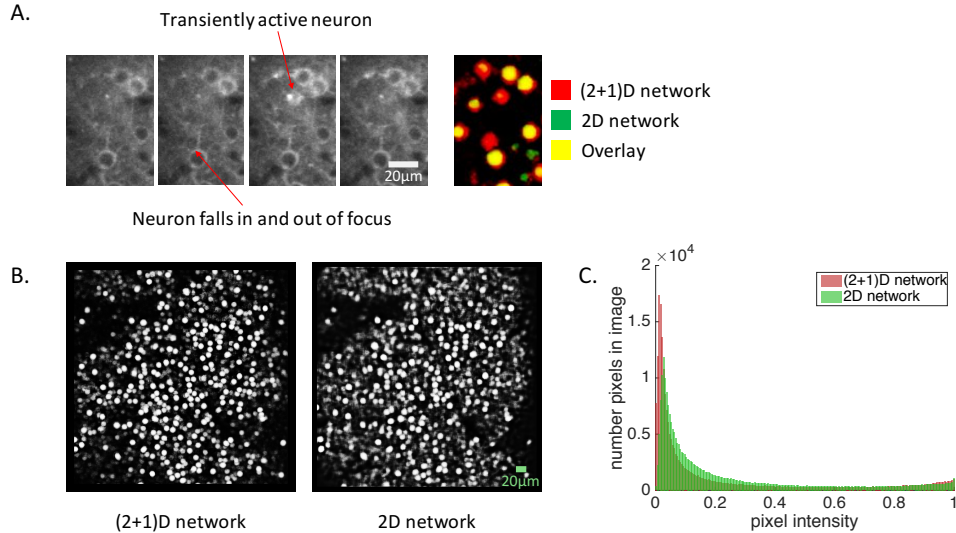

Figure 3: **A**) The (2+1)D network detected neurons that the 2D network failed to locate. The sequence of greyscale images shows a patch of V1 neurons over time. Both transiently active neurons and neurons that wane in and out of the focal plane are visible. The color image shows the output of both networks. The (2+1)D network detects these transiently visible neurons, whereas the 2D network is unable to find these cells using only the mean-flattened image. **B**) The raw outputs of the (2+1)D and 2D networks. **C**) Representative histogram of output pixel intensities. The (2+1)D network output has more values clustered around 0 and 1 compared to the 2D network. This suggests that (2+1)D network output has a higher signal to noise ratio than 2D network output.

ConvNet accuracy was lower on the MEC dataset than the V1 dataset, probably because the former has more noise and larger motion artifacts. The amount of training data for the MEC dataset was also much smaller.

PCA/ICA accuracy was numerically worse, but this result should be interpreted cautiously. PCA/ICA is intended to identify active neurons, while the ground truth included both active and inactive neurons. Furthermore, the ground truth depends on the human expert's selection criteria, which are not accessible to PCA/ICA.

Training and post-processing optimization for ConvNet segmentation took ~6 hours with a forward pass taking ~1.2 seconds per image series. Parameter optimization for PCA/ICA performed by a human expert took ~2.5 hours with a forward pass taking ~40 minutes. This amounted to ~6 hours total computation time for the ConvNet and ~9 hours for the PCA/ICA algorithm. This suggests that ConvNet segmentation is faster than PCA/ICA for all but the smallest datasets.

## 5 Discussion

The lack of quantitative difference between (2+1)D and 2D ConvNet accuracy (same $F_1$ score on the V1 dataset) may be due to limitations of our study, such as imperfect ground truth and temporal downsampling in preprocessing. It may also be because the vast majority of neurons in the V1 dataset are clearly visible in the time-averaged image. We do have qualitative evidence that the (2+1)D architecture may turn out to be superior for other datasets, because its output looks cleaner, and it is able to detect transiently active or transiently in-focus cells (Figure 3).

The (2+1)D ConvNet outperformed PCA/ICA in the precision-recall metrics. We are presently working to compare against recently released basis learning methods [7]. ConvNets readily locate inactive neurons and process new images rapidly once trained. ConvNets adapt to the selection criteria of the neuroscientist if they are implicitly contained in the training set. They do not depend on hand-designed features and so require little expertise in computer vision. ConvNet speed could enable novel applications involving online ROI detection, such as computer-guided single-cell optogenetics [11] or real-time neural feedback experiments.

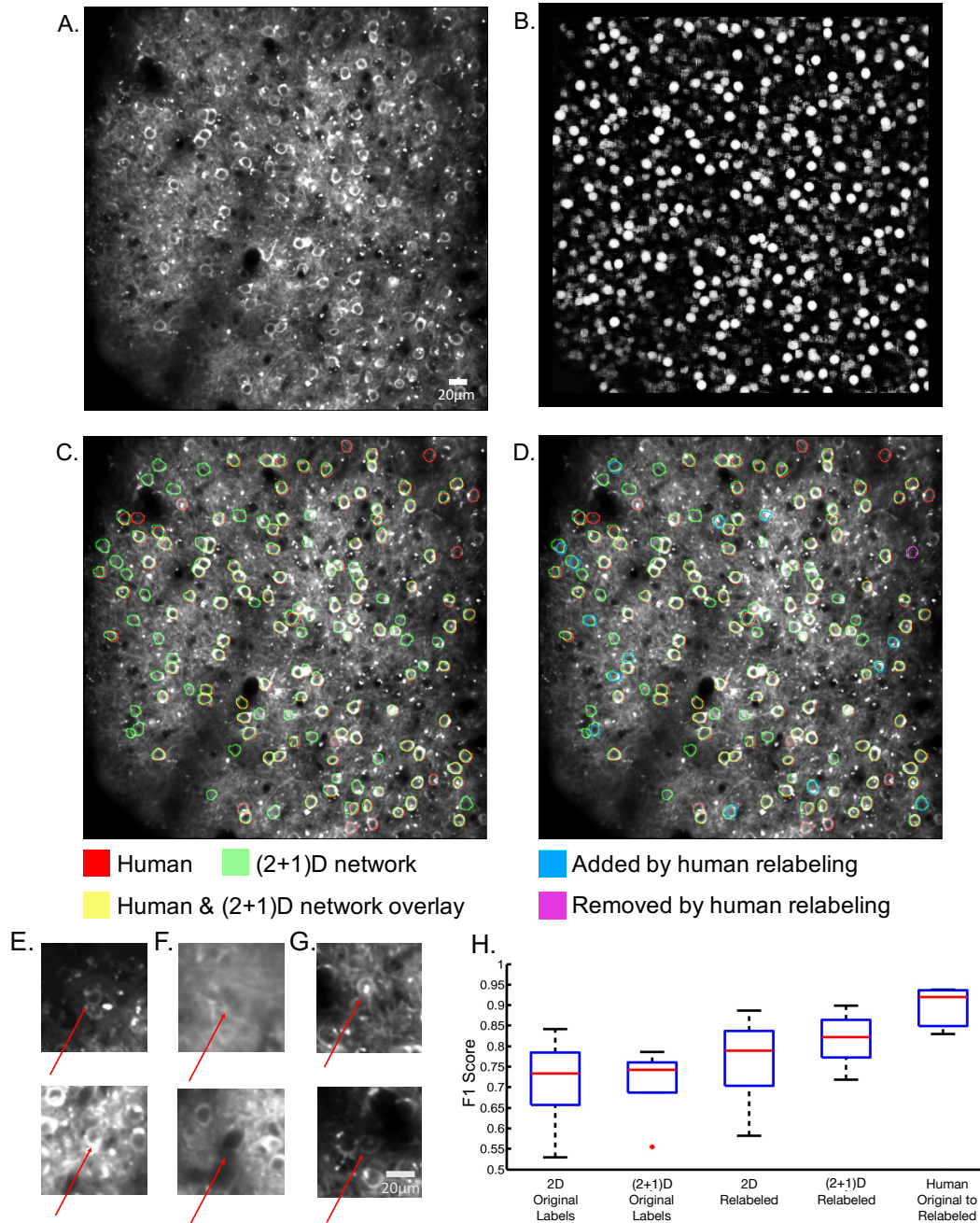

Figure 4: The (2+1)D network successfully detected neurons in the V1 test set with near-human accuracy. **A**) Slice from preprocessed calcium imaging stack input to network. **B**) Network softmax probability output. Brighter regions are considered by the network to have higher probability of being a neuron. **C**) ROIs found by the (2+1)D network after post-processing, overlaid with human labels. Network output is shown by green outlines, whereas human labels are red. Regions of agreement are indicated by yellow overlays. **D**) ROI labels added by human reevaluation are shown in blue. ROI labels removed by reevaluation are shown in magenta. Post hoc assessment of network output revealed a sizable portion of ROIs that were initially missed by human labeling. **E**) Examples of formerly negative ROIs that were reevaluated as positive. **F**) Initial positive labels that were reevaluated to be false. **G**) Examples of ROIs that remained negative even after reevaluation. **H**) $F_1$ scores for (2+1)D and 2D networks before and after ROI reevaluation. Human labels before and after reevaluation were also compared to assess human labeling variability. Boxplots depict the variability of $F_1$ scores around the median score across test images.

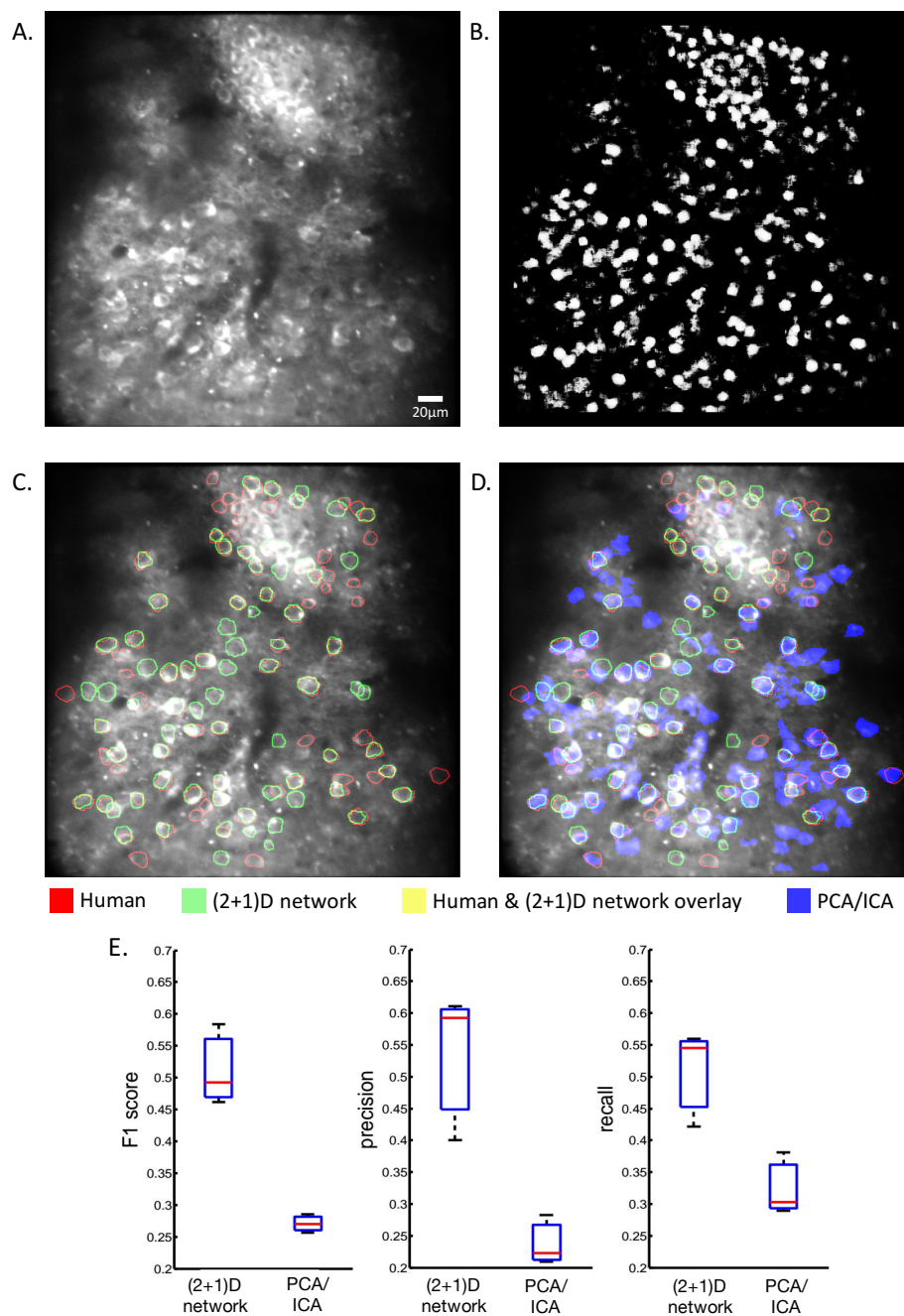

Figure 5: The (2+1)D network successfully detected neurons in the MEC test set with higher precision and recall than PCA/ICA. **A**) Slice from preprocessed calcium imaging stack that was input to network. **B**) Network output, normalized by softmax. **C**) ROIs found by the (2+1)D network after postprocessing, overlaid with ROIs previously labeled by a human. Network output is shown by red outlines, whereas human labels are green. Regions of agreement are indicated by yellow overlays. **D**) The ROIs found by PCA/ICA are overlaid in blue. **E**) Quantitative comparison of $F_1$ score, precision, and recall for (2+1)D network and PCA/ICA on human-labeled MEC data.

**Acknowledgments**

We thank Kisuk Lee, Jingpeng Wu, Nicholas Turner, and Jeffrey Gauthier for technical assistance. We also thank Sue Ann Koay, Niranjani Prasad, Cyril Zhang, and Hussein Nagree for discussions. This work was supported by IARPA D16PC00005 (HSS), the Mathers Foundation (HSS), NIH R01 MH083686 (DWT), NIH U01 NS090541 (DWT, HSS), NIH U01 NS090562 (HSS), Simons Foundation SCGB (DWT), and U.S. Army Research Office W911NF-12-1-0594 (HSS).

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
