[Supplementary Material 1]

# Automatic Neuron Detection in Calcium Imaging Data Using Convolutional Networks

Noah J. Apthorpe, Alexander J. Riordan, Rob E. Aguilar, Jan Homann,
Yi Gu, David W. Tank, H. Sebastian Seung

## Supplementary methods

## Primary visual cortex (V1) calcium imaging data

### Animals

All experiments were performed according to the Guide for the Care and Use of Laboratory Animals (http://www.nap.edu/openbook.php?record_id=12910), and procedures were approved by Princeton University's Animal Care and Use Committee.

### Surgery, behavior, and imaging

Surgical and imaging procedures are described in detail in (1). In short, a viral vector was injected into layer 2/3 of the primary visual cortex of wild type mice (C57/BL6). This viral vector was used to express GCaMP6f, a genetically encoded calcium sensor, in neurons. The vector was delivered to the brain by microinjection performed using a fine fluid-filled glass micropipette. Typical injection volumes were 0.01-0.03 microliters. The injection site was sealed with a thin glass window. A headplate was attached to the skull and the surgical site was carefully sealed. The surgical procedure was performed under general anesthesia.

After a 2/3 week expression period, mice were mounted under a custom-made two-photon excitation microscope for imaging of neural activity. Mice were head-fixed by attaching their headplate to a headpost. Mice were free to run on a large, air-suspended Styrofoam ball placed at a comfortable distance below them. During the imaging sessions, mice were monitored for signs of distress. Visual stimuli were presented to the mouse while brain activity was recorded at cellular resolution with the two-photon microscope setup (512x512 pixel, 30Hz imaging frequency, 16 bits per sample). Samples were taken at about 200-300 micrometer cortical depth. The imaged area was roughly 500x500 micrometers. A typical recording contained about 100-300 neurons and additional active dendrites crossing the imaging plane. Recording sessions lasted for about one hour.

### Data processing

Mouse body movements, especially associated with running on the Styrofoam ball, caused fast lateral frame to frame shifts of the acquired images on the order of 10 micrometers. The roughly 100,000 frames were motion corrected by registering all frames with sub-pixel resolution to a common template by maximizing cross-correlation. The template was constructed by averaging 1000 successive frames. Averaging the frames generated a clear, low-noise reference frame, whereas individual frames were noisy and led to incomplete motion correction. The motion correction procedure was iterated 3 times. Every time a new template was constructed based on the motion-corrected version of the video.

# Medial entorhinal cortex (MEC) calcium imaging data

## Animals
All experiments were performed according to the Guide for the Care and Use of Laboratory Animals (http://www.nap.edu/openbook.php?record_id=12910) and procedures were approved by Princeton University's Animal Care and Use Committee. The Thy1-GCaMP6f-WPRE transgenic mouse line GP5.3 (2), in which layer 2 neurons of MEC were densely labeled by GCaMP6f, was used for all experiments. Mice were generally 9-13 weeks old at the beginning of experiments.

## Surgery, behavior and imaging
All the surgical and imaging procedures were performed as described previously (3). A 1.5mm microprism was surgically inserted into the transverse fissure of the left hemisphere (centered at 3.4mm lateral) to gain optical access to the medial entorhinal cortex. After recovering from the surgery, mice were placed on a water restriction regimen of 1ml water per day. After 2-3 days of water restriction, mice began daily behavioral training (45-60 min) on the virtual linear track at approximately the same time each day. ViRMEn software (4) was used to design, display and control the virtual linear track, which was 1000cm long with textured walls and columns as visual cues. Mice received 4µl water rewards at the beginning and the end of the track. When mice reached the end of the track, they were teleported back to the beginning of the track.

Imaging was performed using a custom, VR-compatible two-photon microscope with moveable objective (3). The microscope consisted of a custom resonance scanner, a 40x water-immersion objective (Olympus, 0.8 NA) and other necessary optics. Ultrasound gel (Sonigel) was used as immersion medial during imaging. A mode-locked Ti:sapphire laser (Chameleon Ultra II, Coherent, 140fs pulses at 80MHz) was used for excitation of GCaMP6f at 920nm. Fluorescence was isolated using bandpass emission filters (542/50nm, Semrock) and detected using GaAsP photomultiplier tubes (1077PA–40, Hamamatsu). Microscope control and image acquisition were performed using ScanImage software (v5.0).

Behavioral imaging experiments began when mice reached a daily performance criterion of 1-2 rewards per minute. For each animal, the microscope's optical axis was aligned with that of the microprism as described previously (3) and the same alignment configuration was used in the following imaging experiments. Calcium dynamics of neurons in layer 2 MMEC were imaged when animals navigated on virtual linear track. For each field of view, images were acquired at 30hz at a resolution of 512x512 pixels (~410x410 µm field-of-view). Each behavioral session lasted 30-40 min.

## Data processing and human ROI labeling
The first 150 frames of each imaging session were removed. The rest of the imaging data was downsampled in time by averaging every three consecutive frames. Motion correction was performed using a whole-frame, cross-correlation-based method (5). All manually-identified regions of interest (ROIs) were selected using Cell Magic Wand or polygon tool in ImageJ (1.47v) (National Institutes of Health) based on eye-inspection of doughnut-like cell body shape.

## ConvNet implementation details

We chose not to use padding or boundary mirroring because several of the microscopy image series had motion correction artifacts at their borders. Thus the network output had an n/2 pixel boundary of 0 probabilities where n is the width of the field-of-view in the spatial dimensions.

## Postprocessing parameter optimization

Thresholding and minimum size postprocessing parameters were optimized independently for the V1 and MEC datasets using reserved validation sets. Pixels with probability values below 0.83 for the V1 dataset and below 0.92 for the MEC dataset were thresholded to 0 in the ConvNet output. Connected components with fewer than 60 pixels for the V1 dataset and fewer than 80 pixels for the MEC dataset were subsequently removed. Watershed regions with fewer than 60 pixels for the V1 dataset and fewer than 80 pixels for the MEC dataset were merged prior to using watershed centroids as seed points for the ImageJ Cell Magic Wand tool. These values are dependent on average neuron size, training time/effectiveness, and optimization grid search granularity and should not be considered "standards" for ConvNet neuron detection.

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

[Supplementary Material 2]



**Supplementary Fig 1.** Precision/recall tradeoff during post-processing, and training loss curves. **A)** Recall vs. precision parameterized over post-processing intensity threshold and minimum neuron size, respectively, for (2+1)D network on V1 dataset. **B)** Pixel classification error across training iterations for (2+1)D network on V1 dataset. **C-D)** Same as A and B, but for MEC dataset. **E)** Pixel classification error across training iterations for 2D network on V1 dataset.