[Reviews · NeurIPS 2016]

Reviewer 1

Summary

The authors establish a framework for training and evaluating a neuron detector for calcium imaging movies. The main goal is to automate ROI annoation and to make this process reproducible. The authors use a windowed convolutional network architecture to produce a score for each pixel indicating the likelihood of belonging to an ROI, then use heuristics to produce a set of ROI centers from this.

Qualitative Assessment

Overall the authors have achieved what they set out to do, and their claims are true. Still, the paper has some major weaknesses that should be at least discussed, and as far as possible addressed before publication. Most importantly, what is this technique really good for? Clearly we're interested in large datasets here, so that means genetically encoded proteins like GCaMPs as synthetics like OGB simply do not scale. But for proteins, is it really enough to detect ROI centers and the use a magic wand tool? The issue of overlapping components is an important one and has been addressed by various factorization techniques as well as even the original GCaMP6 paper. The authors speculate that these techniques could benefit from and ROI detector such as this one, but they should actually show this. For example some factorization code is freely available (https://github.com/epnev/ca_source_extraction), so it's not too much work getting it going. This ca_source_extraction code should absolutely be directly compared to for AP detection as well, as the authors have done for PCA/ICA already. Another related concern is whether this technique will become totally irrelevant in a couple years when someone figures out how to, for example, reliably stain neuronal cytoplasm with a green GECI while staining the nucleus red as well. Another major concern is whether this technique beats off-the-shelf image segmentation/annotation methods that have won competitions in the past. I am not an expert here but I think code should be available for comparisons, and a review of this literature should be included. The problem of circularity in calling data "ground truth" and then modifying it when it doesn't match the output of your algorithm needs a lot more discussion. There might be no way around it, but at the very least the authors absolutely must compare multiple annotators on at least some data and release the raw data and annotations. For widespread adoption, a GPU implementation would be quite helpful. Amazon EC2 is nice to have but prima facie we are dealing with big datasets, so cloud-based computing would not be a first choice. This would also allow less averaging and max pooling in time. The material about "what neuroscientists want" is fairly speculative and should be rewritten with references to actual published studies and their data analysis needs. The authors state that the reason for manual annotation is not clear. I think a major reason, however, is that manual segmentation was used for most validating analyses showing a correlation between electrical and optical signals. The material about the practical aspects of generating ground truth data is excellent. How do the authors propose to deal with zoom factors and pixel sizes that can vary over data? Is there some way of generalizing? The possibility of using a recurrent network to deal with time and/or space should at least be mentioned. The meaning in context of "output patches of size 120x120" is unclear. Why no PCA/ICA in Fig. 1h? Why is performance in MEC so much worse than V1? The mention of "transiently in focus neurons" does not inspire confidence...data with z-drifts should be rejected. The main text should have the values used for thresholding ("low probability", "small components") etc. The process of manual annotation to create ground truth should be describe in more detail. Did users only have the choice of keeping or rejecting the magic wand tool results, or could they perform manual edits too? How much variable was there across users? It's not really clear how the authors define "overlap" in section 2. If one ROI totally contains the other, is the overlap 1? Is overlap symmetric? In fig. 4a-d, the colored lines should be thicker. Inf fig. 4e-g it's not clear what the arrows should point at, centers or edges of ROIS.

Confidence in this Review

2-Confident (read it all; understood it all reasonably well)


Reviewer 2

Summary

This paper describes a classification method for detecting cells in two-photon Calcium imaging recordings. The method is based on convolutional neural networks and uses standard classifiers trained on human-annotated data.

Qualitative Assessment

Following the rebuttal, I decide to keep my scores. I am suspicious of promoting this method, because a majority of the cells visible on the mean image are not active at all, so the method promotes neuropil-dominated ROIs. Fun fact: a new genetically encoded sensor, RCaMP, produces beautiful donuts on the mean image, of which less than 5% have any transients at all. This is a relatively straightforward paper applying established supervised classifier methods (CNNs) to the problem of finding cells in two photon recordings. There appears to be little in the way of algorithmic or computational innovations, therefore this paper should be judged exclusively based on its usefulness and applicability. Unfortunately, a fundamental barrier in the use of any supervised algorithm is the generation of manually annotated data. The authors propose that this be done by each neuroscientist individually, spending tens of hours per each type of dataset. Unfortunately, the statistics of neural recordings are different from brain area to area, between different microscopes and objectives, zoom levels (not a straightforward scaling!), between different Calcium-sensitive sensors, and, importantly, between preparations where only certain cell classes are infected with sensors (i.e. see Allen Institute data). For example, g6f only labels a third of the neurons labelled by g6s, and in ours experience, most cells that appear on the mean image are silent, while active cells do not appear on the mean image as a rule rather than an exception, in densely-labelled recordings where the neuropil is quite bright. As a practically-minded paper, it is worrying that the software requires the use of Amazon services. The authors announce that they will support users of this service, but one has to wonder where the money required to rent such cloud-based services will come from. The implementation could have been made more cost-effective by the use of GPUs, but these are blatantly absent here. It is also worrying that the algorithm is primarily compared to an old method (ICA) which has not been intended for use in cortical recordings, and has not been really updated for use with modern sensors and preparations. It is specified that ConvNets adapt to the selection criteria of the neuroscientist: that seems like a fundamental weakness. A uniformity of preprocessing techniques should be encouraged across neuroscience labs, to help with the reproducibility of results. The authors support the opposite! Minor comments In many places, particularly the introduction, the tone of the authors becomes colloquial. It would be good if these instances were reviewed and rephrased before publication.

Confidence in this Review

3-Expert (read the paper in detail, know the area, quite certain of my opinion)


Reviewer 3

Summary

This paper proposed to use convolution neural network to efficiently detect neurons from the video data of calcium imaging. The method is hopefully to replace the hand-drawing ROI methods in biological image data analysis.

Qualitative Assessment

*After reading the author's rebuttal, I agree them that this method can be combined with CNMF to improve the initialization. I'd like to increase its score of potential impact to 3.* This paper tried to develop a method for automatically analyzing calcium imaging data, which is playing more and more important role in neuroscience research. The proposed method is reasonable and I really enjoy reading it. However, I'm not convinced by its potential impact. First of all, I don't think the results is promising enough to replace the existing manual methods for now. The F1 score for two data sets are just 0.82 and 0.51. For scientific research, this method makes the downstream analysis hard to interpret. Second, even though the method can achieve near-human accuracy, the potential influence of this method is doubtful. In the introduction, they mentioned several unsupervised basis learning method (PCA/ICA, CNMF, dictionary learning, etc. ) and they said the reason for the popularity of manual annotation is not altogether clear. Although the manual method is popular now, replacing it with CNN is not necessarily to be important for the field. Actually unsupervised basis learning methods were proposed recently (2013-2016, PCA/ICA was in 2009 but it has really bad performance) . Although they have superior advantages over the traditional methods, applying them in biological researches requires some computational background, which might be feasible for some experimental labs. Drawing ROI of each neuron, which is proposed in this paper, is a suboptimal method for analyzing calcium imaging method (See the referenced paper 9). It has some problems in extracting cellular temporal activity. Besides these doubts regarding its potential applications, I have have some questions about the analysis in this paper: 1, They picked PCA/ICA to compare their CNN method, but PCA/ICA is detecting independent sources and it depends on the correlation structure of neurons. Each IC is not necessary to be neuron. I would like to see a fair comparison with CNMF or other methods. 2, They used manual results as ground truth and there are some ambiguities in this process. I suggest them trying some datasets with dual labeling of the nuclei.

Confidence in this Review

3-Expert (read the paper in detail, know the area, quite certain of my opinion)


Reviewer 4

Summary

This paper presents the first fully supervised approach to cell detection of two-photon imaging dat, based on a fully convolutional framework (ZNN). The results are compelling and seems not far from the performance of human annotators. Results are compared to a common PCA/ICA approach but not to the state of the art in unsupervised cell detection (e.g., CMNF). The output of this scheme is limited to the spatial domain (unlike other approach which in one complex step determine also spike activity). Performance for awake behaving mice primary visual cortex and medial entorhinal cortex.

Qualitative Assessment

NOTE Following the rebuttal and reading of the other reviews I lowered the "Technical Quality" which is now 2 (originally 3). The authors of Ref1 describe cases where a human annotator could fail to reliably detect and demix cell boundaries. Therefore to claim superiority over such unsupervised methods the current paper needs to compare performance against the performance of such approaches (see Ref1, e.g. using). It is generally required to compare against CMNF in order to claim state of the art accuracy. It is unclear whether the paper used re-evaluation in order to correct the seemingly erroneous ground truth used for annotation. (see specific comment 8 below). It would be generally impressive if the authors can replace or extend the large temporal smoothing and downsampling with network’s kernel that achieve the same result in a supervised manner, e.g., using 3D kernels. If the authors already experienced with 3D training and the results ere not convincing this must be included in the paper. I see this point quite critical in the evaluation of the paper as the temporal smoothing is widely not justified (as I explain in my specific comments below). Specific Comments: 1. 27-28. The authors suggest that CNMF and PCA/ICA are inadequate for some datasets. This point is very weak and incomplete. What are these datasets? And in particular how the current paper is superior over such methods (e.g., CNMF) on these unspecified datasets? 2. 28-29. The authors say that unsupervised methods have not fully replaced the traditional approach. However approaches like CNMF were published in 2016 and other laboratories are likely to share their code in open source pipelines this and next year. Therefore the claim that the traditional approach is still in practice is unjust. 3. 34-39. The text suggests that different experimental setups and scientific goals would require different cell detection outcomes and hence unsupervised approach will depend on parameter tuning. While this may be a true case, it is unclear how much of this burden is actually eliminated by preparation of human ground truth — it seems that all parameter tuning are parametrization of the model, and in the end will require less human effort than ground truth preparation. Moreover, once a laboratory fixed the parameters for several setups the tuning on new datasets (marginal effort) is likely to be greatly reduced. Therefore, I recommend the authors to suppress this paragraph. 4. 39-40. The authors point that current automatic methods require proofreading. However accuracy is what matters — hence also the suggested supervised method as well end up requiring proofreading. Please correct my impression or delete this point. 5. 51. The authors compare to a common method (PCA/ICA) that is not state of the art. First, it should be noted by the authors. Second, it will strengthen the paper if the authors compare to Ref1. 6. 71-77 can be suppressed. 7. 99-101. I find this explanation too heuristic. First at this point the author is unaware of the temporal sampling rate and cannot appreciate the smoothing effect. Second, the usefulness of making the dataset more manageable in size should be marginal where sophisticated methods could perform great (fast and accurate) without data reduction. Also, the procedure used (rectangular filter with max-pooling operation) is nonstandard and require better reasoning. Finally, 132 is read as a drawback of this “Severe reduction.” And then comes 133 which simply goes against the paper’s own method. This should be fixed by introducing better reasoning. 8. 110-115 is confusing. First the V1 dataset is presented. Then a sentence about human annotation time. Then the MEX dataset is presented. Following sentence is about both datasets. And then another seemingly general sentence about human annotation time (I suppose for the MEC dataset since numbers contradict the ones for V1; or could be different estimates of the same human time?). 9. 127 “an an” typo. 10. 132. The choice to heavily down sample in time was the authors’ and hence the explanation in 132 should be revised and made more convincing (the authors can say that other applications were inferior in performance, or come up with a alternative reasoning — perhaps related to the non-isotropic nature of the problem and the different noise distributions in space vs time. 11. 134. Space typo. 12. 136. The large difference of depth between the two datasets should be succinctly explained. 13. 177-179. I strongly disagree with the claim that the reevaluation and the margins reported translate to “nearing human performance.” Instead I suggest the authors repeat this scheme with another human expert and then let the two expert reconcile their disagreements into a mutual call (as done in other fields). It is unfair to let the human be convinced by the algorithm’s output and this is certainly an invalid evaluation metric. Also it is unclear that the reported margin shows near human performance even it this F score was obtained in a valid way. Finally the F1 score of the export w.r.t his own judgment being 0.89 does not represent a standard value for the F score that should be attained. 14. 194-197. Given the above reservations it will contribute if the authors can repeat the “human interrogation” procedure with the PCA/ICA result and see if naive humans are convinced to change their opinion. References: Ref1 ([9] in the submitted paper) Simultaneous Denoising, Deconvolution, and Demixing of Calcium Imaging Data (Neuron 2016)

Confidence in this Review

3-Expert (read the paper in detail, know the area, quite certain of my opinion)


Reviewer 5

Summary

The authors present a convolutional neural network (ConvNet) approach for the problem of detecting cells in large scale calcium imaging recordings. The training procedure consists of manual data annotation that identifies the cell locations and the performance during testing is quantified during the precision/recall framework. The authors describe two different ConvNet architectures. The first (2D+1), applies the network on a sequence of 2D images (roughly the sequence constitutes of downsampled data), that is then max-pooled to produced a single 2D image that quantifies the probability of a cell being present at each pixel. The second architecture (2D) operates on the time averaged data. Both architectures are tested on two different in vivo datasets, coming from the primary visual cortex (V1) and the median entorhinal cortex (MEC). The authors claim that the 2D+1 network can potentially outperform the simpler 2D network and in the case of the MEC dataset it can outperform the PCA/ICA method, a standard automated method used in practice. Moreover, they state that the ConvNet can discover that were not manually annotated, and argue that upon a second round of annotation, their ConvNet approach can approximate human performance.

Qualitative Assessment

As the authors argue the problem of automated ROI selection from calcium imaging recordings becomes increasingly important since these experimental methods grow both in scale and popularity. As a result any automated/semi-automated method to deal with this problem can be helpful. On the other hand ConvNets have recently achieved state of the art performance in many computer vision problems. Since in a given area of the brain neurons have relatively similar shapes, it seems like a natural approach to apply ConvNets for this problem. To my knowledge, the authors are the first to attempt this and I expect more similar methods in the future. For this reason I rated this paper as a potentially high impact one, not necessarily for the general machine learning audience, but definitely for the increasing number of neuroscientists that use this technique. However, I found this work half-baked for several reasons: - As the authors acknowledge, the 2D+1 network fails to consistently give better results than the simpler 2D network that only operates on the average image. That implies to me that the architecture has not been thoroughly explored and the reported results could be possibly improved with more exploration. - The comparison with PCA/ICA is incomplete. As the authors mention PCA/ICA will not detect silent neurons, and therefore the quantitative comparison seems unfair. It would be preferable if the authors could single out the silent neurons from their ground truth (after training) and the results of the ConvNet, and then report the quantitative results again. That shouldn’t be hard to do. - Comparison with other more modern methods (e.g., the CNMF method [9]) is absent. The CNMF methods claims both better computational efficiency and less parameter tuning compared to PCA/ICA so it would be interesting to see whether the ConvNet would still give better performance for the same amount of computational time. - I would imagine that parameter tuning for methods like PCA/ICA and/or CNMF becomes easier the more experienced a user is so reporting the amount of work with respect to time can be misleading (although I cannot think of a different way to do a similar comparison). - The criteria for selecting neurons possibly exclude cases such overlapping cells, and partially observable cells. It would be nice to have some figures (like in Fig 4E-G) that show the ability of the network to detect cells in these cases. - The doughnut shape can be to some extent invariant across different neurons in a given area but the visible dendrites can have much greater variability. It would be nice to test the effectiveness of a ConvNet approach in this case. - Some implementation details are missing: What is the size of the averaging and max-pool windows? Is it equal to the stride? Why is the ConvNet being applied on a 37x37 sliding window instead of the full 512x512 image? Is a different network trained for each window? How is the split in training/validation/test dataset being performed? Is it the same area of the brain but different field of view? It would be helpful to see the different some snapshots of training and testing datasets to get an understanding of how similar these datasets have to be. - In Fig 4D the blue color is hard to distinguish from the green color (at least for me). Better change it with a different color. Based on these comments, I would recommend the authors to work further on testing and optimizing their framework to make their arguments more compelling and their impact wider.

Confidence in this Review

2-Confident (read it all; understood it all reasonably well)